# Enhanced stability of simulated leukocytes for hematology internal quality control samples: A material improvement

**Ngoc Nguyen Vo**⑩*, **Huu Tam Tran***, **Thi Thuy Nhu Le,**
**Thi Hong Phuong Nguyen, Dinh Dung Vu, My Tran Thai, Thi Thanh Phung Nguyen**

Center for Standardization and Quality control in Medical Laboratory of Ho Chi Minh City, Ho Chi Minh City, Vietnam

\* vnnguyen.sdh20@hcmut.edu.vn (NNV); trhuutam@yahoo.com (HTT)

## Abstract

### Objectives

This study aimed to develop and evaluate simulated leukocytes derived from porcine leukocytes as a stable alternative for internal quality control (IQC) in hematology laboratories. Addressing challenges related to material stability and availability, the research contributes to improving laboratory quality assurance in Vietnam.

### Methods

Statistical methods including Shapiro-Wilk test, Levene's test, t-tests, ANOVA, and the IQR method were applied to assess post-production quality and establish target values. Target values were established from data collected across 90 laboratories. The Interquartile Range (IQR) method was used to eliminate outliers, and target values were set for three analyzers (Sysmex XN-1000, Sysmex XN-550, and Horiba ABX Micros 60) per batch at three concentration levels. Sample stability was evaluated over a two-month shelf life using t-tests and monitored for 20 days post-opening with repeated measures ANOVA.

### Results

Simulated leukocytes derived from goose erythrocytes exhibited instability at high concentration levels, whereas those derived from porcine leukocytes maintained stability over two months and demonstrated acceptable performance for up to 10 days post-opening.

### ⛊ OPEN ACCESS

**Data availability statement:** All relevant data are within the manuscript and its Supporting Information files.

**Funding:** The author(s) received no specific funding for this work.

**Competing interests:** The authors have declared that no competing interests exist.

## Conclusions

The findings highlight the potential of porcine leukocytes as a reliable IQC material for hematology, meeting the stability and performance requirements of clinical laboratories.

## 1. Introduction

The implementation of IQC in hematology laboratories is essential for ensuring the accuracy and reliability of diagnostic results, which are critical for patient care and clinical decision-making. According to international guidelines and the national strategy for enhancing laboratory systems, stringent IQC procedures are required in clinical laboratories to meet international standards [1,2]. However, a key challenge faced by laboratories is the limited availability of stable and reliable hematology control materials, particularly those simulating leukocytes, which are crucial for routine IQC processes. Hematology laboratories worldwide encounter difficulties in maintaining an uninterrupted supply of high-quality control materials. This problem often stems from the instability of conventional biological samples used in leukocyte simulation, leading to inconsistencies in test results and increased operational costs. Stabilizing these samples not only ensures the continuity of IQC activities but also promotes technological self-reliance, particularly in developing countries, where dependence on imported control materials poses a significant risk to laboratory efficiency.

Globally, research efforts have focused on developing stable hematology control materials to overcome these challenges. Studies in the UK and Vietnam have demonstrated the importance of reliable IQC samples in minimizing diagnostic errors and improving laboratory performance [3–7]. One of the most persistent challenges in developing hematology control materials is stabilizing leukocyte parameters. Human leukocytes, being highly sensitive to environmental and chemical factors, present significant hurdles in preserving their structural and functional integrity over time. While approaches using fresh human blood showed acceptable WBC simulation, they required analysis within 6–8 hours of collection, limiting their applicability with logistics [7].

Consequently, alternative sources, such as animal-derived simulated leukocytes offer several advantages. Additionally, using animal-derived materials can be more cost-effectivethough but further research is required to determine their efficacy and stability. In this context, goose erythrocytes and porcine leukocytes were considered as promising materials due to their morphological and dimensional similarities to human leukocytes. Goose erythrocytes measure approximately 13.6 µm in diameter and possess a centrally located, oval-shaped nucleus, which facilitates microscopic observation and staining procedures [8]. In contrast, porcine leukocytes comprise all five major white blood cell lineages—neutrophils, eosinophils, basophils, lymphocytes, and monocytes—with diameters ranging from 12 to 13.25 µm and nuclear morphologies closely resembling those of human leukocytes (12–15 µm) [9,10].



Based on these similarities, two separate sample lots using goose erythrocytes and porcine leukocytes were developed to evaluate their potential as substitutes for human leukocytes in IQC materials.

This study aims to evaluate and identify the most suitable material for developing simulated leukocytes in hematology IQC samples. By addressing the challenges of material stability and availability, the research seeks to contribute to the advancement of Vietnam's laboratory quality assurance capabilities, ensuring the reliability of hematology diagnostics in both domestic and international contexts.

## 2. Materials and methods

### 2.1. Materials

**2.1.1. Preparation of simulated human leukocytes. Experiment 1:** Goose blood was collected from geese over 5 months old and stored in sterile 50 mL tubes containing CPDA1 anticoagulant. Whole blood was centrifuged at 2000 rpm for 15 minutes to separate erythrocytes from plasma. The erythrocytes were washed 2–3 times with Phosphate-buffered saline (PBS) buffer to obtain clean cells. Subsequently, the erythrocytes were fixed using a 25% glutaraldehyde-based stabilizing solution containing 0.44 g/l neomycin sulfate. The mixture was gently stirred on a magnetic stirrer for 1 hour to ensure uniform fixation. Fixed erythrocytes were washed 2–3 times with distilled water until the supernatant became clear and preserved in 0.85% sodium chloride solution for future use.

**Experiment 2:** Porcine blood was mixed with a hemolysis solution at a 1:9 (v/v) ratio and acetic acid added to achieve a final concentration of 0.1%. The mixture was incubated at 40°C for 10 minutes. Following incubation, the mixture was transferred into 50 mL centrifuge tubes and centrifuged at 3000 rpm for 5 minutes. The supernatant was discarded to retain the sediment, followed by washing with PBS and repeated centrifugation until the solution became clear, ensuring the removal of all erythrocyte debris.

The sediment (~5 mL) was then suspended in 20 mL of PBS, and a fixation solution was added at a 1:4 volume ratio. The mixture was transferred into a glass container and gently shaken overnight at room temperature. After overnight mixing, the solution was divided into 50 mL centrifuge tubes and centrifuged at 3000 rpm for 5 minutes. The supernatant was discarded to isolate the sediment, which was washed 2-3 times and suspended in 30 mL of PBS for preservation.

The protocol described in this peer-reviewed article is published on protocols.io, https://dx.doi.org/10.17504/protocols.io.n2bvjex8ngk5/v1 and is included for printing purposes as S1 File.

Animal-derived samples were obtained from a licensed facility in Vietnam. The collection, transport, and sample production complied with biosafety regulations and were conducted in a biosafety level 2 laboratory at the Center for Standardization and Quality control in medical Laboratory of Ho Chi Minh City (CSQL of HCMC) Reagents used throughout the study were managed under the ISO 13485:2016—certified quality management system implemented at the research facility, ensuring full traceability and compliance with regulatory and quality assurance standards.

**2.1.2. Ethics statement.** This study involved the use of commercially sourced blood from pigs and geese for the development of hematology internal quality control materials. All blood samples were obtained post-slaughter from licensed commercial suppliers, in accordance with lawful and routine production practices. No live animals were used, handled, euthanized, or subjected to any procedures specifically for research purposes. The study complies with the ethical principles outlined in the ARRIVE guidelines, which emphasize the humane and responsible use of animal-derived materials. The scientific protocol, including the sourcing and use of animal blood, was reviewed and approved by the Institutional Scientific and Technological Committee of the CSQL of HCMC, under the authority of the Ho Chi Minh City Department of Health, Vietnam. Approval was granted under Decision No. 119/QĐ-KCXN.

### 2.2. Method

**2.2.1. Post-production homogeneity assessment.** The samples were assessed for homogeneity post-production using the CellDyn Emerald 22 analyzer to measure white blood cell (WBC) parameters. Each batch (e1, e2) included 14 samples per concentration level for measurement, and all results, including outliers, were considered.

To analyze the data, normality was first tested using the Shapiro-Wilk test due to the small sample size (n<50). A p-value < 0.05 indicated non-normal distribution, while a p-value ≥ 0.05 suggested normality. Depending on the results, different statistical methods were applied.

For normally distributed data, Levene's test (F-test) was used to examine the homogeneity of variances. If variances were equal (p ≥ 0.05), a t-test assuming equal variances was used; if unequal (p < 0.05), a t-test for unequal variances was applied. The hypotheses for the F-test were as follows: $H_0$ assumed equal variances ($\sigma_1^2 = \sigma_2^2$), while $H_1$ indicated unequal variances ($\sigma_1^2 \neq \sigma_2^2$). For the t-test, $H_0$ assumed equal means ($\mu_1 = \mu_2$), whereas $H_1$ suggested different means ($\mu_1 \neq \mu_2$). A p-value < 0.05 led to the rejection of $H_0$, confirming a difference in variances or means, as applicable.

For non-normally distributed data, the Mann-Whitney U test was used to compare the two groups. The process involved ranking all values, calculating rank sums ($R_1$ and $R_2$) for each group, and determining the U statistics using the formula $U_1 = n_1 n_2 + n_1(n_1+1)/2 - R_1$, with $U_2 = n_1 n_2 - U_1$. The smaller of the two U values was then compared against a critical value based on sample sizes and the significance level (α = 0.05). If U was less than or equal to the critical value, $H_0$ (no difference between groups) was rejected. For larger datasets ($n_1, n_2 \geq 20$), z-score approximations were employed for analysis.

This structured approach ensured accurate evaluation of the data, allowing for appropriate statistical conclusions based on distribution and variance characteristics.

The Q-Q plot (quantile-quantile plot) is used to assess whether a dataset follows a theoretical distribution by plotting its quantiles against the expected quantiles of the distribution; a linear pattern indicates good agreement, while deviations suggest departures from the theoretical model.

**2.2.2. Outlier elimination and target value determination.** The experimental sample sets, comprising three concentrations (level 1, level 2, level 3) with initial values measured by the CellDyn Emerald 22, were distributed for hematology analysis on representative analyzers to obtain the target values for each analyzer model.

**Experiment 1:** Samples were sent to 60 laboratories using different types of analyzers, and results were obtained from 51 participating laboratories, include: Horiba ABX Micros 60 (16 labs), Sysmex XN-1000 (22 labs), and Sysmex XN-550 (13 labs).

**Experiment 2:** Samples were sent to 60 laboratories using various analyzers, and results were obtained from 39 participating laboratories, include: Horiba ABX Micros 60 (13 labs), Sysmex XN-1000 (16 labs), and Sysmex XN-550 (10 labs).

Outliers, defined as values significantly divergent from the dataset, were excluded before calculating summary statistics, including mean (X), standard deviation (SD), coefficient of variation (CV), and control limits. The Interquartile Range (IQR) method was employed, with boundaries set at the upper limit (Q3+1.5×IQR) and the lower limit (Q1−1.5×IQR), a widely accepted threshold that balances sensitivity to deviations while minimizing the excessive exclusion of valid data. This approach ensures robust data integrity and enhances the reliability of summary statistics and control limits, which are critical for evaluating the accuracy and precision of hematology analyzers. Target values were computed as X±4SD after outlier elimination [11].

**2.2.3. Post-opening stability evaluation.** WBC parameters were monitored daily over 20 consecutive days under storage conditions of 2–8°C. Post-opening stability was assessed for two batches (from Experiments 1 and 2) using four analyzers: Sysmex XN-550 (machine1), Sysmex XN-1000 (machine2), Horiba ABX Micros 60 (machine3), and Cell-Dyn Emerald 22 (machine4).

Repeated Measures ANOVA was employed to analyze daily changes in WBC parameters. The analysis considered time (a within-subjects factor with 20 days), concentration level (a between-subjects factor), and analyzer type (a covariate). The significance level was set at 0.05, with 95% confidence intervals. A p-value (Sig.) less than 0.05 indicated significant changes over time, suggesting instability.

Mauchly's test for sphericity, typically used to evaluate the equality of variances for repeated measures, was not a primary concern in this analysis. Since the main factors included time and concentration level, and analyzer type was treated as a covariate, the assumption of sphericity was less critical. Additionally, the dataset was reviewed to ensure that any violations of sphericity would not significantly impact the results. If necessary, adjustments such as Greenhouse-Geisser correction were applied to maintain the validity of the ANOVA outcomes.

**2.2.4. Shelf-life stability assessment.** Stability at two months was evaluated using the Cell-Dyn Emerald 22. Three sample sets were tested for each batch at the two-month expiry. An initial baseline measurement, recorded immediately after production, was used as a reference value. A one-sample t-test for variance was conducted using SPSS ($\alpha = 0.05$) to compare the results over time.

- **p-value ≤ α:** Reject $H_0$, indicating instability.

- **p-value > α:** Fail to reject $H_0$, indicating stability.

## 3. Results

### 3.1. Post-production homogeneity assessment

As shown in Table 1, the coefficients of variation (CV%) across all levels and batches ranged from 2.19% to 7.41%, which are well below the allowable total error (TEa) of 15.49% for white blood cell (WBC) parameters as recommended by Westgard [12]. This indicates acceptable variability in the post-production results. The Shapiro-Wilk test p-values were all above 0.05, suggesting that the data followed a normal distribution. Levene's test for equality of variances also returned p-values greater than 0.05 across all levels, indicating homogeneity of variances between batches. Furthermore, the t-tests for equality of means showed no statistically significant differences between batches at any level (all p > 0.05), supporting the consistency and reproducibility of the sample production process.

Fig 1 illustrate the degree to which the observed data aligns with a normal distribution. In each plot, the data points predominantly fall along the reference line, indicating a reasonable approximation to normality. However, noticeable deviations occur in the plots for e1, l2 and e2, l2, particularly at the higher extremes, suggesting the presence of outliers or slight skewness. Overall, while the datasets generally exhibit normality, careful consideration of the observed deviations is essential for robust statistical interpretation.

### 3.2. Outlier elimination and target value determination

Table 2 presents the descriptive statistics of processed data used for outlier elimination and target value determination. The coefficient of variation (CV%) across all instruments and concentration levels remained below the allowable total error (TEa) of 15.49% for white blood cell (WBC) parameters, as recommended by Westgard, indicating acceptable analytical precision. In the table, $N_0$ represents the total number of results obtained, while N indicates the number of results retained

**Table 1. Distribution and Homogeneity assessment analysis of post-production results.**

| Batch | Level | CV% | p-value | | |
|-------|-------|-----|---------|---|---|
| | | | **Shapiro-Wilk test** | **Levene Test for Equality of Variances** | **t-test for Equality of Means** |
| e1 | 1 | 7.41 | 0.82 | 0.89 | 0.77 |
| e2 | 1 | 6.01 | 0.53 | 0.60 | 0.64 |
| e1 | 2 | 3.99 | 0.59 | 0.12 | 0.32 |
| e2 | 2 | 2.19 | 0.13 | 0.82 | 1.00 |
| e1 | 3 | 3.19 | 0.66 | 0.89 | 0.83 |
| e2 | 3 | 3.83 | 0.77 | 0.50 | 0.76 |



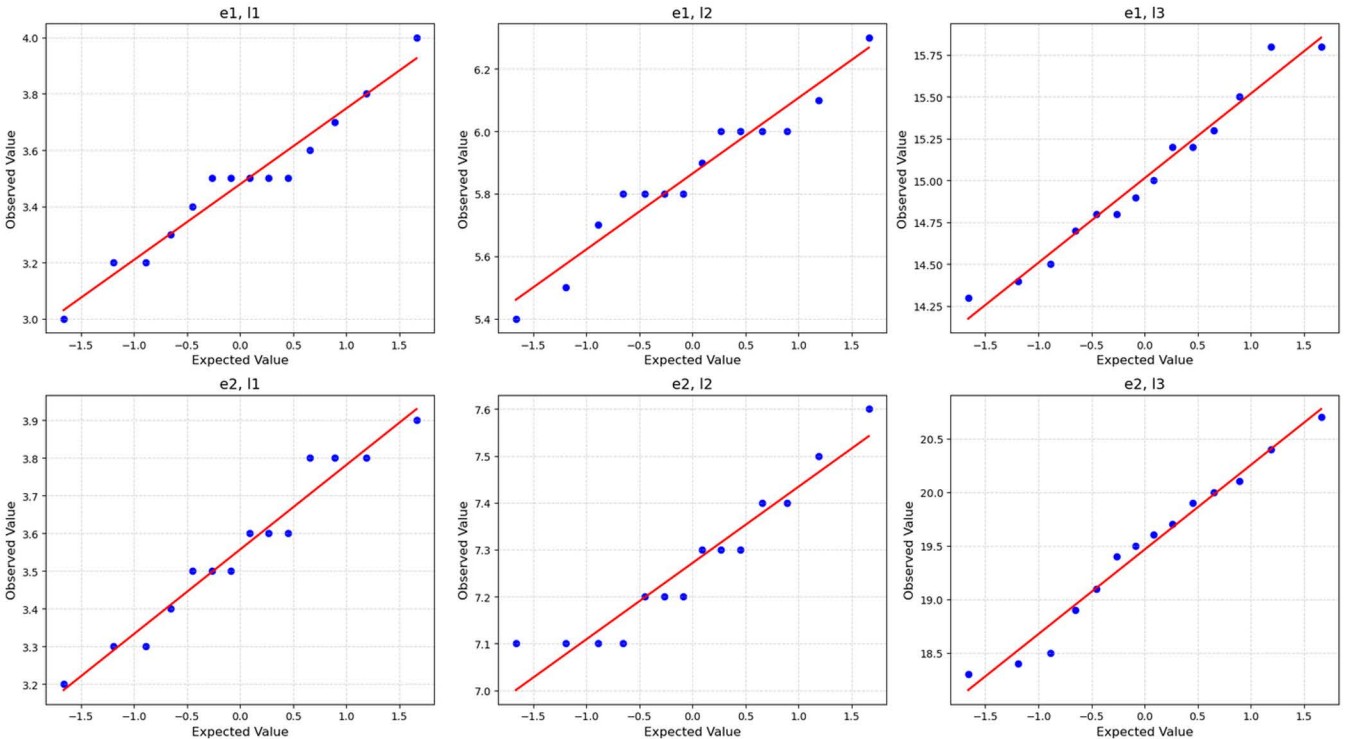

**Fig 1. Q-Q Plot of sample batches post-production.**

**Table 2. Descriptive statistics of samples by instruments, batches, and concentration levels.**

| Machine | Batch | Level | Percentiles | | | Lower Limit | Upper Limit | Number of result | | Median (X) | SD | CV (%) |
|---|---|---|---|---|---|---|---|---|---|---|---|---|
| | | | Q1 | Q3 | IQR | | | $N_0$ | N | | | |
| Horiba ABX Micros 60 | e1 | 1 | 4.63 | 5.20 | 0.58 | 3.76 | 6.06 | 16 | 16 | 4.90 | 0.44 | 8.90 |
| | | 2 | 9.08 | 9.75 | 0.68 | 8.06 | 10.76 | 16 | 16 | 9.40 | 0.50 | 5.28 |
| | | 3 | 25.08 | 27.83 | 2.75 | 20.95 | 31.95 | 16 | 16 | 26.30 | 2.04 | 7.77 |
| | e2 | 1 | 5.03 | 5.90 | 0.87 | 3.73 | 7.21 | 13 | 13 | 5.70 | 0.57 | 10.07 |
| | | 2 | 9.50 | 11.10 | 1.60 | 7.10 | 13.50 | 13 | 13 | 10.30 | 1.13 | 10.95 |
| | | 3 | 26.10 | 29.80 | 3.70 | 20.55 | 35.35 | 13 | 13 | 28.00 | 2.76 | 9.87 |
| Sysmex XN-1000 | e1 | 1 | 4.26 | 4.62 | 0.36 | 3.72 | 5.16 | 22 | 19 | 4.43 | 0.32 | 7.26 |
| | | 2 | 7.45 | 8.65 | 1.20 | 5.65 | 10.44 | 22 | 20 | 7.74 | 0.82 | 10.62 |
| | | 3 | 23.36 | 25.55 | 2.19 | 20.07 | 28.83 | 22 | 22 | 24.57 | 1.44 | 5.85 |
| | e2 | 1 | 3.98 | 4.28 | 0.31 | 3.52 | 4.74 | 16 | 14 | 4.10 | 0.21 | 5.15 |
| | | 2 | 6.78 | 7.55 | 0.77 | 5.63 | 8.70 | 16 | 16 | 7.13 | 0.62 | 8.75 |
| | | 3 | 18.46 | 20.93 | 2.47 | 14.77 | 24.63 | 16 | 15 | 19.09 | 1.82 | 9.55 |
| Sysmex XN-550 | e1 | 1 | 6.54 | 6.92 | 0.38 | 5.97 | 7.49 | 13 | 10 | 6.75 | 0.29 | 4.34 |
| | | 2 | 12.21 | 15.02 | 2.81 | 8.00 | 19.24 | 13 | 13 | 14.16 | 2.13 | 15.05 |
| | | 3 | 29.74 | 33.01 | 3.27 | 24.84 | 37.92 | 13 | 11 | 29.98 | 2.73 | 9.10 |
| | e2 | 1 | 5.21 | 5.53 | 0.31 | 4.74 | 5.99 | 10 | 10 | 5.45 | 0.27 | 4.98 |
| | | 2 | 8.49 | 8.97 | 0.47 | 7.78 | 9.67 | 10 | 9 | 8.77 | 0.42 | 4.79 |
| | | 3 | 22.09 | 24.27 | 2.18 | 18.82 | 27.54 | 10 | 9 | 22.63 | 1.56 | 6.87 |

after outlier removal, which were used for calculating the target value. A comparison between batches shows that the number of excluded results in batch e2 was generally lower than in batch e1, implying that test results using batch e1 samples yielded more outliers during the statistical screening process.

Fig 2 presents a boxplot summary of measurement distributions by instrument, sample batch, and concentration level. The plots reflect the spread and symmetry of results, supporting the descriptive statistics shown in Table 3. Several box-plots—especially at Level 3—exhibit extended whiskers, indicating broader variability. For instance, batch e2 at Level 3 on the Horiba ABX Micros 60 showed an interquartile range (IQR) of 2.75 (Q1 = 25.08, Q3 = 27.83), while batch e2 reached an even wider IQR of 3.70 (Q1 = 26.10, Q3 = 29.80), with long whiskers extending toward both lower and upper limits (20.95–31.95 and 20.55–35.35, respectively). Similarly, Level 3 samples from batch e2 on Sysmex XN-550 also displayed an IQR of 2.18 with whiskers covering a broader range (18.82–27.54), suggesting higher dispersion at this level.

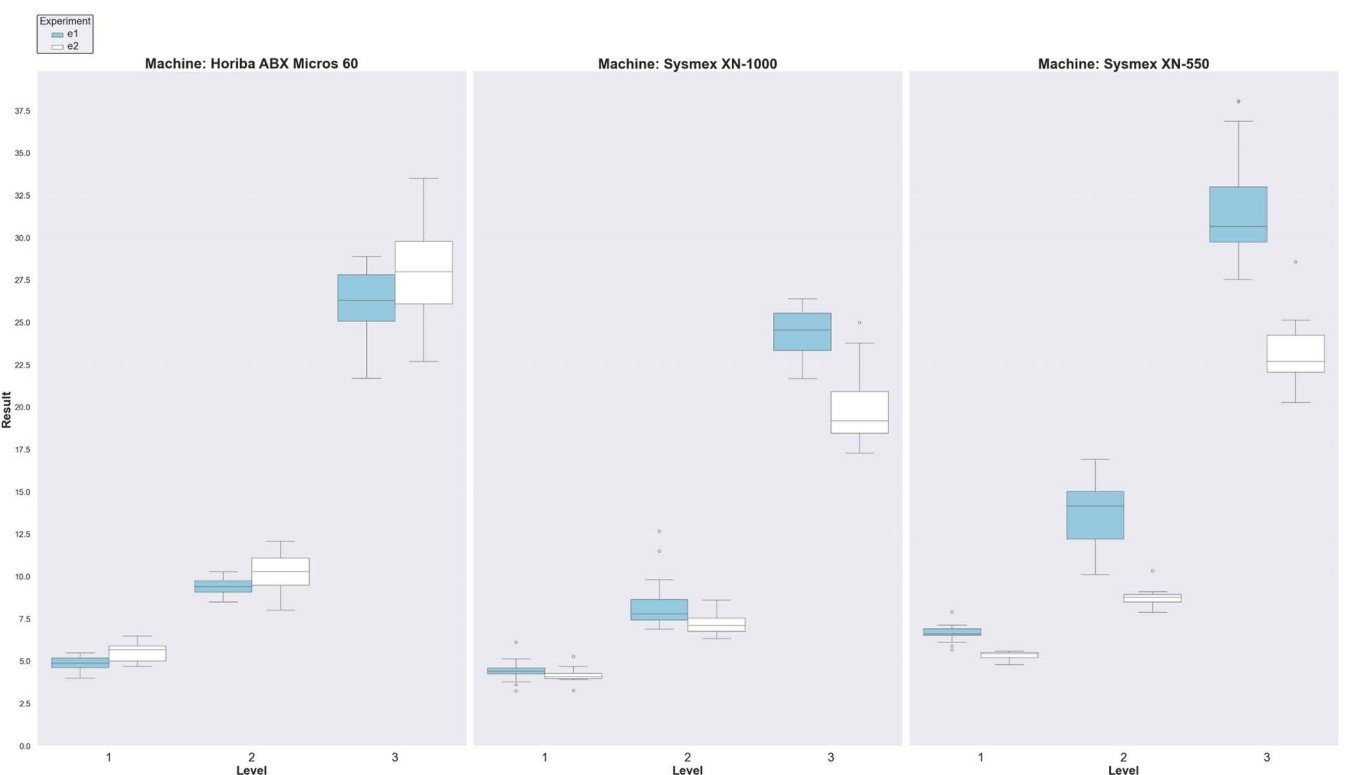

**Fig 2. Boxplot summary showing statistics of samples by instruments, experiments, and concentration levels.**

**Table 3. Repeated measures ANOVA results.**

| Factor/Interaction | Batch 1 | | | | Batch 2 | | | |
|---|---|---|---|---|---|---|---|---|
| | df | F | p-value (Sig.) | Partial Eta Squared ($\eta_p^2$) | df | F | p-value (Sig.) | Partial Eta Squared ($\eta_p^2$) |
| day | 19 | 2.392 | 0.002 | 0.230 | 19 | 1.333 | 0.182 | 0.210 |
| day * machine | 19 | 1.671 | 0.047 | 0.173 | 19 | 0.771 | 0.735 | 0.134 |
| day * level | 38 | 1.330 | 0.117 | 0.249 | 38 | 1.184 | 0.253 | 0.321 |

In contrast, some boxplots demonstrated tighter distributions, reflecting good homogeneity and repeatability. For example, batch e2 at Level 2 on Sysmex XN-550 had a narrow IQR of 0.47 (Q1 = 8.49, Q3 = 8.97) with no extreme outliers, and a low CV% of 4.79%, indicating stable measurements. Similarly, batch e2 at Level 2 on Horiba ABX Micros 60 also showed a compact distribution (IQR = 1.60) with moderate whisker length and no significant deviations.

While a few outliers were observed, they were limited in number and were statistically managed during outlier elimination, as reflected in the $N_0$ and N values in Table 3. These visual patterns complement the statistical analysis, reinforcing the suitability of the samples across batches and concentration levels for use in internal quality control validation.

### 3.3. Post-opening stability evaluation: Repeated measures ANOVA

The results in Table 3 demonstrate significant findings, with a p-value of 0.002 for the 'day' variable, indicating notable differences in hematology parameters over time and contributing to the assessment of stability for Batch 1. Notably, the interaction between 'day' and 'machine' also showed significance, with a p-value of 0.047, suggesting that the type of analyzer influences the stability changes over time. The p-value for the interaction between 'day' and 'level' was 0.117, implying that concentration levels might not affect sample properties statistically significant within this framework.

In contrast, the results of Batch 2 showed no statistically significant differences between days for the second batch, with a p-value of 0.182 for the 'day' variable, exceeding the significance threshold of 0.05. This indicates that hematology parameters remained relatively stable over the 20-day observation period. Furthermore, the interaction between 'day' and 'machine' yielded a p-value of 0.475, suggesting no significant impact of analyzer type on sample stability. Similarly, the interaction between 'day' and 'level' had a p-value of 0.253, providing no evidence to support a significant effect of concentration levels on the post-opening stability of the samples.

Fig 3 shows that Batch 1 presented some challenges in terms of stability. Specifically, Horiba ABX Micros 60 at concentration level 2 produced measurements outside the target range from day 1 onward (Table 3) and failed to return to acceptable values throughout the 20-day period. Similarly, Sysmex XN-1000 and Sysmex XN-550 at concentration level 3 recorded non-compliant results, reflecting significant instability during this timeframe. In contrast, concentration level 1 exhibited relatively stable results across all machines, suggesting that improvements may be needed to enhance the stability of samples at higher concentration levels.

Fig 4 shows that Batch 2 demonstrated overall stability across all machines and concentration levels during the first 10 days post-opening, with values consistently falling within the target range (Table 3). On the Sysmex XN-1000, Level 1 remained stable throughout the 20-day period, while Level 2 showed instability from day 11 and Level 3 slightly exceeded the target range on days 19 and 20. On the Sysmex XN-550, all three levels stayed within the target range until day 9, with out-of-range results observed from day 10 (Level 2), day 13 (Level 1), and day 17 (Level 3). On the Horiba ABX Micros 60, Levels 1 and 2 remained consistently stable, and Level 3 showed a single out-of-range value on day 15.

### 3.4. Shelf-life stability assessment

The results from Table 4 indicate significant variability among the samples at the expiry. For Batch 1, concentration level 2 was stable with a p-value 0.065, while levels 1 and 3 were unstable, showing p-values <0.05. In contrast, Batch 2 exhibited stability across all concentration levels, with p-values of 0.300, 0.081, and 0.800, indicating that these samples maintained their integrity well beyond the two-month period.

## 4. Discussion

This study demonstrates the successful development and evaluation of simulated leukocytes derived from porcine leukocytes, which exhibited enhanced stability compared to goose erythrocyte-derived materials. The findings are supported by rigorous statistical analyses, including normality testing, homogeneity of variances, and inter-batch mean comparisons.

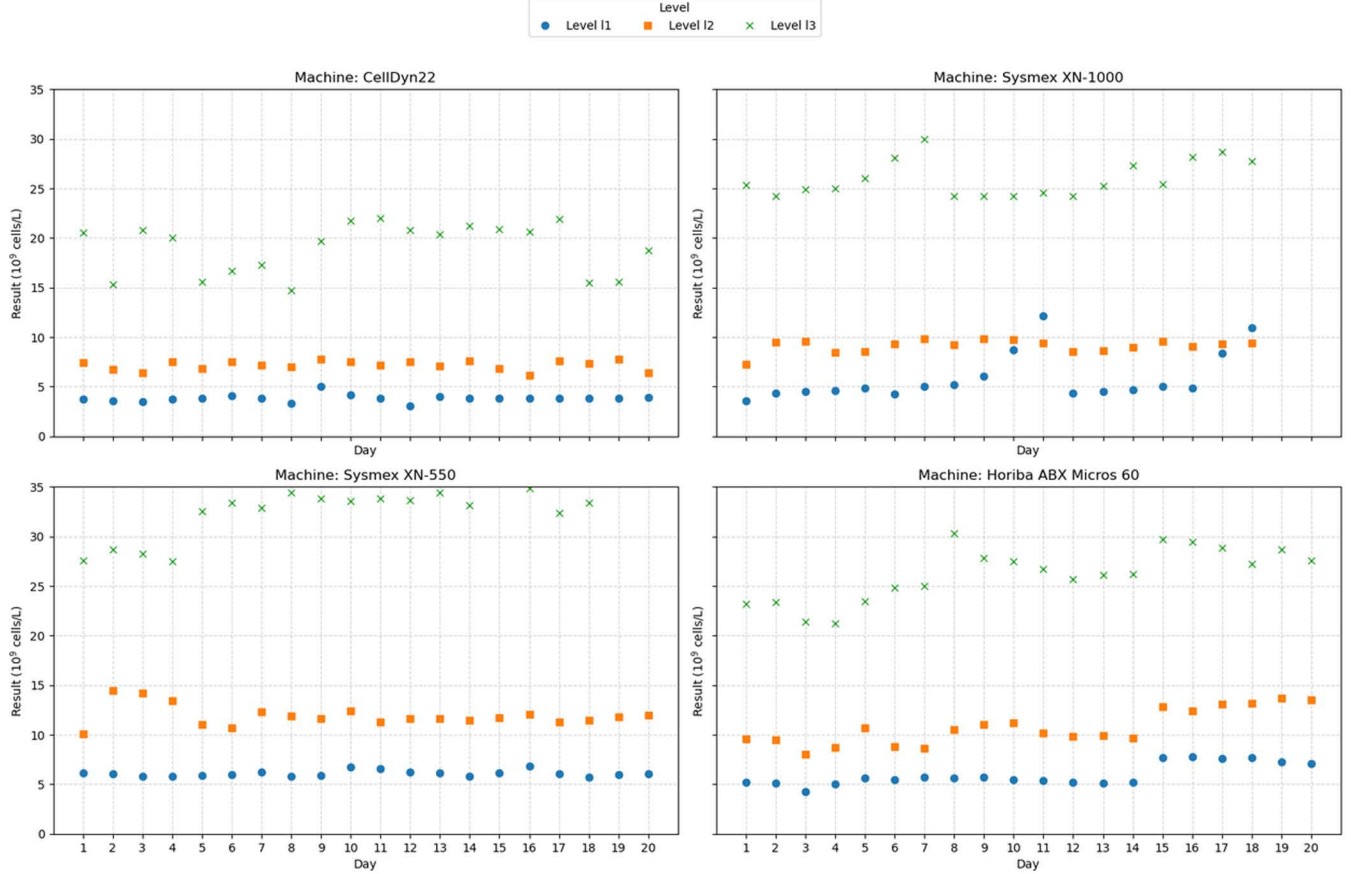

**Fig 3. The results of 20-day post-opening stability monitoring for Batch 1 across analyzer types.**

Post-production homogeneity results confirmed that both experimental batches (e1 and e2) were statistically consistent, with CV% values well below the allowable total error (TEa) threshold of 15.49% for WBC parameters as recommended by Westgard.

The selection of alternative animal-derived sources in this study was based on their morphological resemblance in size and optical properties to human leukocytes, making them suitable materials for simulating WBC in hematology quality control materials. The improved performance of batch e2 (porcine-derived) over batch e1 (goose-derived) may also be explained by practical and biological factors. Blood collection from a single pig yields a considerably larger volume than from a goose, enabling the preparation of a batch from fewer donor animals. This reduces inter-individual biological variation, such as differences in the membrane stability, fixation response, or cell morphology. In contrast, collecting sufficient goose blood for one batch typically requires pooling from multiple animals, increasing the potential for cell heterogeneity and batch instability. These variations may have contributed to the greater data dispersion and reduced stability observed in the goose-derived samples during post-opening evaluations.

Regarding the selection and number of participating laboratories, the initial study design aimed to include 120 laboratories—60 receiving batch e1 and 60 receiving batch e2. In practice, valid data were received from 51 laboratories for the goose group and 39 for the porcine group, totaling 90 laboratories. Although slightly below the initial plan, the sample size remains statistically adequate. With 90 participants (51 vs. 39), the recalculated statistical power is approximately 75% for

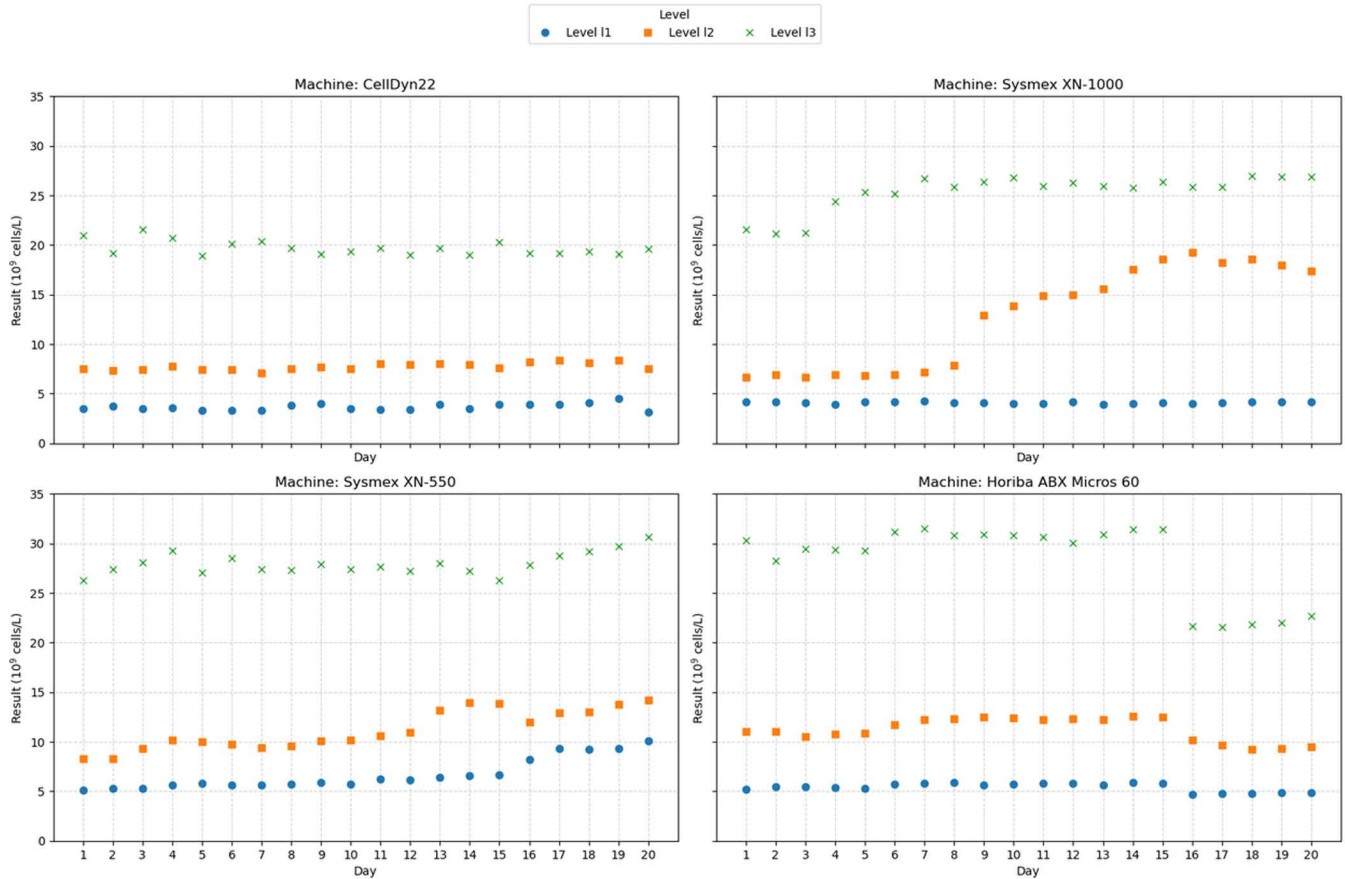

**Fig 4. The results of 20-day post-opening stability monitoring for Batch 2 across analyzer types.**

**Table 4. Shelf-life stability assessment results.**

| Batch | Concentration level | Original value | p-value two-side (sig.) | Conclusion |
|---|---|---|---|---|
| 1 | 1 | 4.28 | 0.006 | Unstable |
| 1 | 2 | 7.79 | 0.065 | Stable |
| 1 | 3 | 20.55 | 0.001 | Unstable |
| 2 | 1 | 4.28 | 0.300 | Stable |
| 2 | 2 | 7.79 | 0.081 | Stable |
| 2 | 3 | 20.55 | 0.800 | Stable |

detecting a standardized mean difference of 0.6 at a 5% significance level, as noted in the revised Methods section. This adjustment ensures transparency and supports the validity of the study's conclusions.

To address inter-laboratory variability, the study applied an outlier elimination procedure using the interquartile range (IQR) method. This approach provided a robust and objective means of excluding aberrant data points, ensuring that the derived target values accurately reflect the central tendency of valid measurements across diverse instruments and laboratory settings. Boxplot visualizations in Fig 2 reinforced the statistical findings, with most distributions showing acceptable spread and symmetry. While certain subgroups—particularly those at higher concentrations—exhibited extended whiskers and wider IQRs (e.g., batch e2 at Level 3), the overall data dispersion remained within acceptable limits. Notably, some



concentration levels and batches (e.g., batch e2 at Level 2 on Sysmex XN-550) demonstrated tight, stable distributions with minimal outliers, reflecting consistent material performance.

Stability evaluation was conducted in two stages: shelf-life stability at two months and post-opening stability over 20 days. Results indicated that the porcine leukocyte-derived samples (batch e2) were stable across all concentration levels and analyzers during both periods, while the goose-based samples (batch e1) showed signs of instability at levels 1 and 3. It should be acknowledged that shelf-life testing was conducted using only one analyzer, which is a limitation of the current study. This constraint was due to limited resources. Nevertheless, the stability protocol and formulation applied here are based on a patented research solution developed by the CSQL of HCMC. The patented solution—awarded at the 27th Ho Chi Minh City Technical Innovation Contest—demonstrated that a mixture of goose erythrocytes could be used to simulate leukocytes, forming a complete and stable whole blood control sample. That foundational work has supported the routine application of this formulation in EQA production at CSQL of HCMC since 2022. Since its initial launch in 2022 with 65 laboratories, the hematology EQA program utilizing this formulation has expanded significantly, reaching 168 participating laboratories by 2025. The program distributes two stabilized samples per round, six times per year, with two-month shelf-life. No quality failures have been reported since implementation, further reinforcing the reliability of the formulation evaluated in the present study. While stability testing across multiple analyzers will be considered in future studies, previous datasets provide initial evidence supporting the decision to conduct stability testing on a single hematology analyzer.

Despite the promising results, limitations were identified, including the reliance on a single laboratory for stability assessments, potential influences from storage conditions, equipment, and sample handling practices. These factors may contribute to deviations from expected post-opening stability, emphasizing the importance of standardizing laboratory protocols to ensure consistent performance of IQC materials. Additionally, we acknowledge that the statistical tools applied in this study were used to assess data consistency and temporal variation, and do not in themselves constitute clinical validation of sample stability.

## 5. Conclusions

This methodology offers significant contributions to the broader field of hematology quality control by presenting a practical, cost-effective, and reproducible approach to simulating leukocytes in hematology IQC materials. By leveraging locally available animal-derived materials, the study provides a viable alternative to fresh human samples, which are often logistically challenging and limited by shelf-life. The established two-month shelf-life and the demonstrated stability for up to 10 days post-opening represent notable improvements over previous methodologies. While the results indicate that no significant degradation observed over the 10-day post-opening period, further research is necessary to address the limitations identified, including the influence of laboratory conditions and sample handling. This work lays the groundwork for future advancements in hematology diagnostics and quality control, contributing to the self-reliance of Vietnamese laboratories in producing reliable and stable control materials.

## Supporting information

**S1 File.  A protocol for producing quality control materials for hematology testing using porcine and goose blood.**
(PDF)

**S2 File.  Hematology IQC data.**
(XLSX)

## Acknowledgments

We extend our gratitude to the CSQL of HCMC for providing the time and facilities necessary for this study.



## Author contributions

**Conceptualization:** Ngoc Nguyen Vo, Huu Tam Tran.

**Investigation:** Ngoc Nguyen Vo, Huu Tam Tran.

**Methodology:** Ngoc Nguyen Vo, Dinh Dung Vu, My Tran Thai.

**Supervision:** Huu Tam Tran, Thi Thuy Nhu Le.

**Validation:** Thi Hong Phuong Nguyen.

**Visualization:** Thi Thanh Phung Nguyen.

**Writing – original draft:** Dinh Dung Vu, My Tran Thai.

**Writing – review & editing:** Ngoc Nguyen Vo, Thi Hong Phuong Nguyen.

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
