## [Decision Letter · Decision Letter 0]

11 Jun 2025

PONE-D-25-24409Enhanced stability of simulated leukocytes for Hematology Internal quality control samples: A material improvementPLOS ONE

Dear Dr. Vo,

Thank you for submitting your manuscript to PLOS ONE. After careful consideration, we feel that it has merit but does not fully meet PLOS ONE’s publication criteria as it currently stands. Therefore, we invite you to submit a revised version of the manuscript that addresses the points raised during the review process.

We look forward to receiving your revised manuscript.

Kind regards,

Sarman Singh, MD, FRSC, FRCP

Academic Editor

PLOS ONE

Journal Requirements:

3. In the online submission form, you indicated that the data that support the findings of this study are not publicly available in order to respect the confidentiality agreements established with participating clinical laboratories and to protect proprietary information related to specific hematology analyzer models. As part of the study protocol, participating laboratories were assured that their individual results would be coded and accessed only under controlled conditions.

Researchers who are interested in accessing the data for academic purposes may contact the corresponding author. Data will be made available upon reasonable request and under terms that ensure the privacy and confidentiality of the participating laboratories are maintained.

4. Please update your submission to use the PLOS LaTeX template. The template and more information on our requirements for LaTeX submissions can be found at http://journals.plos.org/plosone/s/latex.

5. Please ensure that you refer to Figure 3, and 4, in your text as, if accepted, production will need this reference to link the reader to the figure.

6. We note you have included a table to which you do not refer in the text of your manuscript. Please ensure that you refer to Table 6 in your text; if accepted, production will need this reference to link the reader to the Table.

Additional Editor Comments :

Conclusion section can be shortened.

Reviewers' comments:

Reviewer's Responses to Questions

**Comments to the Author**

1. Does the manuscript report a protocol which is of utility to the research community and adds value to the published literature?

Reviewer #1: Yes

Reviewer #2: Yes

2. Has the protocol been described in sufficient detail?

To answer this question, please click the link to protocols.io in the Materials and Methods section of the manuscript (if a link has been provided) or consult the step-by-step protocol in the Supporting Information files.

The step-by-step protocol should contain sufficient detail for another researcher to be able to reproduce all experiments and analyses.

Reviewer #1: Yes

Reviewer #2: Yes

3. Does the protocol describe a validated method?

Reviewer #1: No

Reviewer #2: Yes

4. If the manuscript contains new data, have the authors made this data fully available?

Reviewer #1: N/A

Reviewer #2: Yes

**5. Is the article presented in an intelligible fashion and written in standard English?**

Reviewer #1: Yes

Reviewer #2: Yes

6. Review Comments to the Author

Reviewer #1: General Evaluation:

The manuscript addresses an important and practical issue in clinical laboratory science: Improving the stability of internal quality control (IQC) materials in hematology. The study explores an Innovative and Resource-conscious approach by using porcine erythrocytes as a base material for simulating leukocytes, offering a viable solution to challenge faced particularly in developing countries. The work is well-structured and methodologically sound good, and the findings are relevant to laboratories aiming to improve local quality assurance capabilities.

However, there are several areas where the manuscript could be strengthened to enhance clarity, scientific rigor, and global applicability.

Major Comments:

1. Novelty and Relevance:

The study contributes a valuable regional solution with broader applicability, especially in low- and middle-income settings. However, the manuscript would benefit from a clearer articulation of how this approach advances or differs from previous studies that have investigated animal-derived IQC materials.

2. Methodological Rigor:

The statistical methods are generally appropriate and well explained. However, more detail is needed regarding the criteria for selecting the 90 laboratories and how inter-laboratory variability was controlled or accounted for.

In the stability testing, only one laboratory was used for shelf-life analysis. While the authors acknowledged this limitation, additional data or a plan for multi-laboratory validation in future studies should be discussed more explicitly.

3. Comparison Between Materials:

The comparison between goose and porcine erythrocytes is central to the study but lacks depth in the discussion. A more thorough biochemical or structural rationale for the observed differences in stability would strengthen the conclusions.

4. Data Transparency:

The data availability statement restricts public access due to confidentiality concerns. While this is understandable, it limits reproducibility. Consider depositing anonymized aggregate datasets or offering more extensive supplementary data.

Minor Comments:

1. Language and Style:

The manuscript is mostly well written, but several grammatical and typographical errors remain . A thorough language and style edit is recommended to enhance readability.

2. Figure and Table Presentation:

Figures and tables are informative, but some (e.g. boxplots and Q-Q plots) would benefit from clearer axis labeling and legends. Ensure all figures are fully self-explanatory.

3. Ethics Statement:

The ethics review process is appropriately addressed, but a brief note on the absence of zoonotic risk or biosafety measures in handling porcine blood would be helpful.

Recommendations:

Accept with major revisions: The study provides valuable insight, but several key clarifications and enhancements are needed before publication. In particular, the authors should:

• Expand discussion on the scientific basis of porcine erythrocyte superiority.

• Include a more robust strategy for future multi-laboratory validation.

• Improve language and data presentation.

Reviewer #2: 1. Overall Reviewer Summary

This manuscript describes a novel approach for developing simulated leukocytes from porcine erythrocytes to serve as internal quality control (IQC) material for hematology analyzers. The methodology is clearly presented and the statistical analyses are well executed. The study addresses a relevant need for stable, cost-effective IQC materials, particularly in resource-limited settings.

The study demonstrates technical rigor and real potential impact; however, some claims—particularly those related to validation and stability—should be moderated to reflect the limitations of single-lab assessments and absence of direct comparisons with commercial controls. The discussion would benefit from further context regarding clinical utility, analytical acceptability limits, and broader reproducibility.

2. Validity of the methodology and analyses

The methodology is generally sound and described in sufficient detail. The authors have used appropriate statistical tests (Shapiro-Wilk, Levene’s test, t-tests, ANOVA, etc.) to assess homogeneity and stability over time. However:

• The authors should clarify that statistical significance does not confirm analytical or clinical stability.

• While the IQR method for outlier removal is valid, the definition of acceptable analytical variation (e.g., %CV thresholds) should be stated more explicitly.

• The use of a single laboratory for shelf-life testing is a limitation that must be emphasized more directly in the methods and discussion.

3. Quality of writing and presentation

The manuscript is generally well organized and readable. The writing is clear, though certain sections could benefit from stronger scientific tone:

• A few grammar issues are present (e.g., “storage condition” should be “storage conditions”) and should be corrected.

• The abstract effectively summarizes the content, though use of terms like “comprehensive statistical approach” could be more precise.

4. Significance of findings and conclusions

The study addresses a significant issue—access to stable IQC materials—and provides a novel, potentially cost-effective solution using porcine erythrocytes. However:

• The findings need to be interpreted with greater caution.

• Authors should elaborate on practical implementation: cost, biosafety, scalability.

• A comparative discussion with existing commercial IQC materials would greatly strengthen the paper.

5. Specific Comments to Authors

Major Comments

• Clarify that statistical tools were used to assess sample consistency, not to “validate” stability as a clinical outcome.

• Discuss the lack of reference to commercial IQC products and how your material compares in terms of performance or acceptability (e.g., ±15% tolerance or allowable total error).

• Address the limitation of performing stability testing in a single laboratory and outline steps for future multi-site validation.

• Include more commentary on biosafety, reagent sourcing, and practical implementation.

• Highlight how this methodology contributes to the broader field of hematology QC or other diagnostic settings.

Minor Comments

• Line 84: The t-test hypotheses contain a typo: it says σ22 twice instead of σ1² and σ2².

• Line 131: Typo — “An one-sample t-test” → “A one-sample t-test.”

• Line 208: “Successful development” → consider rephrasing to “development and evaluation.”

• Line 225: “Storage condition” → “storage conditions.”

• Line 232: “Time does not significantly affect stability” → “no significant degradation observed over the 10-day post-opening period”.

7. PLOS authors have the option to publish the peer review history of their article (what does this mean? ). If published, this will include your full peer review and any attached files.

**Do you want your identity to be public for this peer review?** For information about this choice, including consent withdrawal, please see our Privacy Policy .

Reviewer #1: No

Reviewer #2: **Yes: ** Duressa Shafi Ahmed

---

## [Author Response · Author response to Decision Letter 1]

23 Jul 2025

The revised manuscript has been carefully updated in accordance with all comments and suggestions provided by the reviewers and the academic editor. We sincerely appreciate the valuable feedback and guidance, which have significantly improved the quality of the manuscript. We hope that the current version meets the journal’s standards and look forward to your further evaluation.

---

## [Editor Report · Decision Letter 1]

30 Jul 2025

Enhanced stability of simulated leukocytes for Hematology internal quality control samples: A material improvement

PONE-D-25-24409R1

Dear Dr. Vo,

We’re pleased to inform you that your manuscript has been judged scientifically suitable for publication and will be formally accepted for publication once it meets all outstanding technical requirements.

Kind regards,

Sarman Singh, MD, FRSC, FRCP

Academic Editor

PLOS ONE

Additional Editor Comments (optional):

Congratulation

---

## [Editor Report · Acceptance letter]

PONE-D-25-24409R1

PLOS ONE

Dear Dr. Vo,

I'm pleased to inform you that your manuscript has been deemed suitable for publication in PLOS ONE. Congratulations! Your manuscript is now being handed over to our production team.

Kind regards,

on behalf of

Professor Sarman Singh

Academic Editor

PLOS ONE